# Multiple Intra-Articular Injections of Adipose-Derived Mesenchymal Stem Cells for Canine Osteoarthritis Treatment

**DOI:** 10.3390/cells14050323

**Published:** 2025-02-20

**Authors:** Xianqiang Li, Xuwei Jian, Ziyin Yan, Huazhen Liu, Lisheng Zhang

**Affiliations:** College of Veterinary Medicine/Bio-Medical Center, Huazhong Agricultural University, Wuhan 430070, China; xianqiangli@webmail.hzau.edu.cn (X.L.); xwjian1225@outlook.com (X.J.); yzyreus@webmail.hzau.edu.cn (Z.Y.); lhz219@mail.hzau.edu.cn (H.L.)

**Keywords:** osteoarthritis, mesenchymal stem cells, MSC therapy, cartilage repair

## Abstract

Osteoarthritis (OA) is one of the most common degenerative diseases in dogs and humans, which can lead to articular cartilage deterioration, chronic pain, and decreased quality of life. The anti-inflammatory, anti-fibrotic, analgesic, and cartilage regeneration properties of mesenchymal stem cell (MSC) therapy provide a new direction for the treatment development of OA in the future. Currently, MSC therapy lacks confirmed ideal sources, dosages, formulations, and specific characteristics. In this study, we evaluated the efficacy of multiple canine adipose-derived mesenchymal stem cell (ADSC) injections on anti-inflammation and joint cartilage damage in a canine OA model. Considering animal ethics, we simulated the effects of inflammation and cartilage repair during treatment through a mouse OA model. In the mouse OA model, through the detection of cartilage repair and inflammation-related key factors via histology and molecular biology, it was found that MSC therapy has a certain repair effect on cartilage, but the anti-inflammatory effect is time-dependent. In the canine OA model, we verified the feasibility of multiple injections of ADSCs. Compared with the control group, the cartilage repair effect of the treatment group was obvious, and the inflammatory factors decreased, showing an obvious therapeutic effect. This study demonstrates that multiple intra-articular injections of canine ADSCs could be effective in treating OA symptoms.

## 1. Introduction

Osteoarthritis (OA) represents a prevalent degenerative disease in human and animal populations, including canines, leading to the degradation of joint cartilage integrity, chronic pain, reduced quality of life, and increased mortality [1,2]. The Global Disease Report estimates that more than 527 million people suffer from OA, which places a significant personal and socioeconomic burden on individuals due to its impact on exercise capacity [3,4]. In the UK alone, over 200,000 new cases of canine OA are diagnosed annually, with age, obesity, and injury being the primary etiological factors [1,5,6]. Due to the low regenerative capacity of the cartilage matrix, the onset of this disease is often irreversible [7]. Despite decades of research, current therapeutic approaches remain largely symptomatic, relying on non-steroidal anti-inflammatory drugs, hyaluronic acid injections [8], or surgical interventions in severe cases [9]. However, no treatment has been completely successful in restoring hyaline cartilage [10]. Only a revolutionary change in the way OA patients are treated can radically reduce the negative effects of osteoarthritis.

Mesenchymal stem cell (MSC) therapy has emerged as a promising therapeutic candidate for OA treatment, offering anti-inflammatory, anti-fibrotic, analgesic, and chondrogenic properties [11]. Among various MSC sources, including bone marrow [12], adipose [13], dental pulp [14], placenta [15], and synovium [16], adipose tissue stands out as the preferred source due to its minimally invasive accessibility and abundance [17]. MSC therapy has demonstrated potential in preventing and reversing cartilage degradation across multiple species models [18,19,20,21], with direct intra-articular administration showing significant therapeutic efficacy [22,23].

During regeneration, inflammation often precedes actual repair of the damage, attempting to restore homeostasis after acute injury, with successful tissue regeneration attributed to a controlled inflammatory process mediated by pro-inflammatory cytokines like the interleukin-6 (IL-6) family [24]. However, aberrant and prolonged activation of IL-6 cytokines is related to various fibrotic pathologies [25] and the pathogenesis of complex chronic diseases such as OA, where progressive inflammation and degenerative changes in the articular cartilage are accompanied by excessive fibrous proliferation and collagen production in the synovial tissue and subchondral bone [26].

However, MSC therapy faces certain limitations, particularly regarding cell persistence post-injection, with most cells surviving less than 48 h [27,28]. While single injections of allogeneic adipose-derived mesenchymal stem cells (ADSCs) have shown promising results in canine OA [22,29], the chronic nature of OA necessitates investigation into multiple injection protocols [30]. Although MSCs exhibit immune-privileged properties [31], the immune response varies with the cell type, administration route, and restimulation timing, requiring specific evaluation of each therapeutic protocol.

Clinical data in humans have confirmed the long-term safety of MSCs in orthopedics, with a low incidence of adverse events [32]. Although MSCs have some immune evasion [31], the immune response largely depends on the cell type, route of administration, and timing of restimulation, and each treatment plan should be specifically evaluated. Since there are no relevant studies on dogs, it is necessary to evaluate the effects of multiple injections and periodic assessments before widely promoting MSC therapy in the pet market. Studies have shown that MSC therapy can undergo xenogeneic treatment [33], and it has been applied in the treatment of OA, promoting bone and cartilage regeneration without immune rejection. Xenogeneic MSCs do not trigger cellular immune activation in short-term treatments and can be immunotolerant in the host environment [34], suggesting that xenogeneic models may be an effective strategy for elucidating the preliminary mechanisms of MSC therapy, considering animal ethics.

This study aims to evaluate the therapeutic efficacy of multiple canine ADSC administrations in knee OA treatment. We first characterized isolated canine ADSCs, followed by assessment of their anti-inflammatory and chondroprotective effects in a murine knee OA model. Subsequently, we investigated the feasibility and efficacy of repeated ADSC administrations in a canine knee OA model, with comprehensive monitoring of therapeutic outcomes throughout the treatment period.

## 2. Materials and Methods

### 2.1. Detection of Morphology and Proliferative Capacity of Canine ADSCs

P3 ADSCs at 90% confluence were digested and counted, seeded into 96-well culture plates at a density of 1 × 10^4^ cells/well, and cultured at 37 °C in a 5% CO_2_ incubator. The next day, the original culture medium was discarded, and 100 μL of new culture medium and 10 μL of CCK-8 working solution were added. After 2 h of incubation in the incubator, the absorbance of each well was measured at 450 nm using a microplate reader and recorded, and then a growth curve was plotted.

### 2.2. Isolation of Adipose-Derived Stem Cells

Under sterile conditions, fat from the groin of a 1-year-old Beagle dog weighing approximately 5 kg was isolated. The tissue was washed three times with phosphate buffered saline (PBS), cut into small pieces, and incubated in Collp solution (1% Collagenase, Type 1) at 37 °C for 4 h. After digestion, the tissue was filtered through a 100 μm cell sieve to remove undigested tissue fragments, and the obtained filtrate was centrifuged at 500× *g* for 10 min. The resulting cells were resuspended in low-glucose DMEM (DMEM-LG) with 10% fetal bovine serum (FBS). ADSCs were seeded at a density of 10^6^ cells/mL in Dulbecco’s Modified Eagle Medium containing 1.0 g/L glucose (DMEM-LG; Gibco, Thermo Fisher, Beijing, China), 10% MSC certified fetal bovine serum (FBS), 100 U/mL penicillin, and 100 μg/mL streptomycin (cytiva, Thermo Fisher, China). Cells were cultured at 37 °C, 5% CO_2_, and 85% humidity. The medium was changed every two days until 70% confluence was reached.

### 2.3. Multipotent Differentiation Detection

OriCell^®^ mesenchymal stem cell differentiation induction kits (Cyagen Biosciences, Suzhou, China) were used to identify different differentiation potentials. First, 1 mL of 0.1% gelatin was added to a six-well plate, and after being placed in a sterile environment for 30 min, the gelatin was removed. Second-generation ADSCs were seeded at a density of 5.0 × 10^3^ cells/cm^2^ in a six-well culture plate and cultured in control medium for 24 h. Then, the medium was replaced according to differentiation targets with osteogenic, adipogenic, or chondrogenic medium. The cell medium was changed every 3 days, and after 21 days of continuous culture, Alizarin Red, Oil Red O, and Alcian Blue staining were performed on the differently differentiated cells. Cell morphology was observed and recorded every 7 days during the induction process.

### 2.4. Preparation and Treatment of Mouse OA Model

Ten-month-old female C57 mice underwent destabilization of the medial meniscus (DMM) to construct a mouse osteoarthritis model. The mice were anesthetized, the right hind limb knee joint was shaved (about 2–3 mm) and alternately disinfected with alcohol and povidone-iodine, and a 1–1.5 cm incision was made. The skin and subcutaneous fascia were separated. The joint capsule was opened between the patellar ligament and the medial collateral ligament, the knee was flexed, surrounding fat tissue was cleared, and the medial meniscotibial ligament was exposed. The meniscus was transected under a stereomicroscope (Olympus Corporation, Tokyo, Japan), the knee was flushed with saline, the patella was repositioned, and the joint capsule and skin incisions were sutured. Arthritis symptoms gradually appeared 6 weeks post-surgery. The SHAM group opened the joint capsule but did not transect the medial meniscotibial ligament, directly suturing the joint capsule. Treatment began 8 weeks after instrumentation of the mice, with each group’s mouse weight recorded at the start and then every 2 weeks for a total of 4 times. Weight changes were plotted against time on the x axis, and the mouse weight was plotted on the y axis. A caliper was used to measure and record the instrumented knee joint width and thickness at the same time intervals, plotted as knee swelling changes over time. we transected the medial meniscus and tibial collateral ligament surgically, initiated the first ADSC treatment 8 weeks after modeling, administered a second ADSC treatment 2 weeks later, and switched to PBS in the fourth week to detect changes in indicators in mice after stopping ADSC injections. The experiment was divided into an SHAM group, DMM group, and DMM + ACs group. All mice were randomly assigned, with 18 rats in each group. Six rats were euthanized and sampled at each time point. Each mouse was injected with 10 μL of the cell suspension (1 × 10^7^ cells/mL).

### 2.5. Preparation and Treatment of Canine OA Model

Eight-month-old healthy female Chinese garden dogs were prepared to construct a canine osteoarthritis model. Canine knee osteoarthritis (KOA) modeling was performed via cranial cruciate ligamentectomy (CrCLt) surgery, simulating cartilage damage caused by ligament rupture and forming mild knee osteoarthritis. The dog’s hind limb knee joint area was shaved 10 cm above and below for preparation, and a venous catheter placed in the dog’s cephalic vein. SuTai (5 mg/kg) was injected intravenously for anesthesia and connected to an anesthesia machine, and basic signs such as breathing and heart rate were monitored via a stethoscope. Saline was injected for rehydration, and surgery commenced once anesthesia was achieved. In lateral recumbency, an incision was made from the tibial tuberosity through the skin on the medial side of the patellar ligament. The joint capsule was opened along the skin incision. The patella was displaced laterally, the fat pad retracted distally, exposing the cruciate ligament and meniscus, and the cruciate ligament was cut, showing a positive drawer sign. Absorbable sutures were used to layer the joint capsule, muscle layer, and skin. Post-surgery, daily intravenous infusion (Cefoperazone Sulbactam Sodium, 20 mg/kg) continued for 5 days to prevent infection. From 10 days post-surgery, a daily walk of 1–2 h was observed in the dog’s walking gait for 8 weeks. X-ray examinations (55 kVp, 6.3 mAs, FFD: 100 cm) were performed before instrumentation of the dogs, before stem cell treatment, and before sampling to assess knee joint modeling and treatment effects. The walking state and joint discomfort of each dog were evaluated every day during the treatment. The degree of pain in the knee joint could be judged by whether the animal’s legs had retraction reactions or whether there was an uncomfortable cry through continuous palpation of the injured knee joint. Images were collected during sampling. Following confirmation of diagnosis, injections were performed every 2 weeks, totaling three injections. Samples were taken in the eighth week after treatment, and the dogs were euthanized before sampling. The experiment was divided into a KOA + PBS group and KOA + ADSCs group, with a total of 6 dogs randomly divided into 3 dogs in each group. Each canine was injected with 1 mL of a cell suspension (1 × 10^7^ cells/mL).

### 2.6. Histological Analysis

Local cartilage tissue was taken from the abraded healthy joints in mice and dogs, the cartilage and bone were removed together with bone forceps, and the tissue was placed in formaldehyde fixative for 24 h and then in decalcifying solution for 1–2 h. Damaged joints were collected from animal hind limbs, with their hair trimmed and muscle removed before being fixed for 48 h in 4% paraformaldehyde and then placed in 10% EDTA for decalcification for 30 days. Decalcified samples were embedded in paraffin and sectioned into 5 μm slices. HE staining, Toluidine Blue staining, and Safranin O-Fast Green staining kits were used to stain the sections. Stained tissue sections were observed under an inverted microscope (Olympus Corporation, Tokyo, Japan).

### 2.7. Scratch Assay

Local cartilage tissue was obtained from the abraded-healthy junction of the dog, and a number of superficial cartilage sheets were meticulously scraped using a scalpel blade. Chondrocytes were then obtained by means of an enzymatic digestion process. Lines were drawn evenly on the back of a six-well plate using a ruler every 0.5 cm, with at least 5 lines crossing each well. Chondrocytes were plated into the six-well plate at a density of 5 × 10^5^ cells per well, and after changing the medium the next day, scratches were made on the cells with a 10 μL pipette tip aligned with the lines. After scratching, PBS was used to wash the wells, with medium added for culture, and placed in a 37 °C 5% CO_2_ incubator. Samples were taken at 0 h, 6 h, 24 h, 48 h, and 72 h. Specific regions were observed for cell migration using an inverted microscope and photographed. Image Pro Plus 6.0 was used to randomly select 6–8 horizontal lines to calculate the average distance between cells.

### 2.8. Immunofluorescence

Immunofluorescence staining was used to assess the expression of specific surface markers CD34, CD45, CD90, and CD105 in the ADSCs and canine chondrocyte collagen type II (Col-II) immunofluorescence identification. Cells were seeded at a density of 5.0 × 10^3^ cells/cm^2^ in cell culture plates with slides until 50% confluence was reached. The cells were washed twice with PBS and fixed with 4% paraformaldehyde at room temperature for 30 min. After washing three times with PBS, the residual liquid was removed with absorbent paper, and 3% goat serum was applied to the slides for blocking at room temperature for 30 min. After removing the blocking solution, sufficient diluted primary antibody was added to each slide and incubated in a humidified chamber at 37 °C for 4 h. PBS washes were performed 3 times, each for 3 min, and excess liquid was removed with absorbent paper. Then, diluted fluorescent secondary antibody was added, followed by humidified chamber incubation at 37 °C for 1 h. Tris-buffered saline with Tween-20 (TBST) and PBS washes were performed, followed by 4′,6-diamidino-2-phenylindole (DAPI) incubation in the dark for 5 min. Excess liquid was removed, and the slides were mounted with antifade reagent-containing mountant before imaging under a fluorescence microscope (CarlZeiss Microscopy, LSM710, Jena, Germany).

### 2.9. Statistical Analysis

GraphPad Prism 7.03 software (GraphPad Software, San Diego, CA, USA) was used for statistical analysis. The measurement data were expressed as the mean ± standard error of the mean (mean ± SEM). Statistical significance between groups was determined by unpaired student’s *t*-tests. A value of *p* < 0.05 was considered statistically significant.

## 3. Results

### 3.1. Characteristic Identification of Canine ADSCs

Figure 1 demonstrates the characteristics of canine ADSCs in terms of morphological features, growth status, differentiation ability, and surface markers. Under the microscope, it was observed that approximately 48 h after cell isolation and culture, adherent growth occurred. The cells appeared to be round, triangular, or spindle-shaped. After media changes, the proliferation rate increased, and the proliferating cells arranged themselves in a vortex or radial pattern, showing strong cell polarity characteristics, with no significant differences observed until the fourth passage (Figure 1A). The ADSCs at the fifth passage still maintained a stable morphology and growth rate. The CCK8 results showed that the third-passage ADSCs began to proliferate rapidly 24 h after passaging, entering the logarithmic growth phase and reaching a plateau at 96 h. The overall growth curve of the cells presented a typical “S” shape, indicating good proliferation capability (Figure 1B).

To assess the multipotent differentiation potential of ADSCs, the third-passage ADSCs were cultured and stained using different induction media. On the 7th day of osteogenic induction, calcium salt deposition was observed locally; on the 14th day, mineralized nodules were stained red with Alizarin Red; and on the 21st day, a large number of red mineralized nodules were observed. During adipogenic induction, the cells became shorter and rounder at day 7, with translucent lipid droplets forming locally. The Oil Red O staining turned red on the 14th day, and after staining on the 21st day, some lipid droplets were lost, while the remaining ones appeared red. During chondrogenic differentiation, the cell morphology changed to irregular long triangles by the 7th day; with Alcian Blue staining, the cells appeared blue-green by the 14th day; and by the 21st day, a large number of cells exhibited blue-green coloration (Figure 1C). This demonstrated the multipotent differentiation capability of ADSCs. Immunofluorescence analysis showed that ADSCs were negatively expressed for surface markers CD34 and CD45 and positively expressed for CD90 and CD105 (Figure 1D).

Based on the fibroblast-like morphology, proliferation capacity, multipotent differentiation potential, and surface marker expression of the cultured cells in vitro which aligned with the standards set by the International Society for Cellular Therapy for stem cell identification, it is confirmed that the cells extracted from canine inguinal fat were high-purity, well-characterized ADSCs.

### 3.2. Comparison of the Therapeutic Effects of ADSCs at Different Time Points on Mouse Osteoarthritis

At the start of the treatment, the body weight of the SHAM group mice was significantly higher than that of the model group (*p* < 0.01). During the ADSC treatment, the body weight of the mice in the DMM + ADSCs group increased steadily with better growth conditions, showing significant differences from the DMM group at the 4th week (*p* < 0.05) but no notable difference from the SHAM group. After the treatment ceased, the mice’s body weight decreased, and by the 6th week, there was no significant difference in body weight among the three groups (Figure 2C). The kneecaps’ width and thickness visibly showed swelling in the joints post-modeling, with an upward trend over time. The DMM group consistently differed from the SHAM group during treatment (*p* < 0.001). Improvement in swelling was observed in the DMM + ADSCs group between 0 and 4 weeks, with significant differences from the DMM group (*p* < 0.001) and showing no distinction from the SHAM group by the fourth week, indicating relatively healthy knee joints. However, swelling was significant by the 6th week after stem cell treatment cessation, with a striking difference from the SHAM group (*p* < 0.001) and noticeable swelling (Figure 2D).

Since joint cartilage damage is the main histopathological characteristic of OA joints, evaluation using HE, Safranin O-fast green staining provides a significant reference value for ADSCs’ effects on joint cartilage damage over different periods. After treatment in the DMM group, HE staining of the knee joint tissues showed smooth cartilage surfaces, a lack of synovial tissue hyperplasia, abundant chondrocytes, an undamaged cartilage matrix, and clear tide lines in the SHAM group. In contrast, the DMM and DMM + ADSCs groups exhibited rough cartilage surfaces, significant synovial tissue hyperplasia, thickened cell layers, and severe destruction of the cartilage matrix, confirming successful model construction. Despite the discovery of irregular cartilage surfaces and cartilage degeneration in the DMM + ADSCs group, ADSC treatment reduced the morphological changes and degeneration of joint cartilage compared with the DMM group. Damage in the DMM group worsened over time, with extensive cartilage destruction and loss, osteophyte formation, and an increase in inflammatory factors (Figure 2E). Safranin O-fast green staining showed smooth, intact cartilage tissue and no fibrosis on the surface for the SHAM group mice. In contrast, both the DMM injury group and the DMM + ADSCs treatment group displayed severe cartilage degeneration, uneven cartilage surfaces, disordered chondrocyte arrangements, and other phenotypes. Over time, damage in the DMM group intensified, with distinct cartilage damage and fissures, uneven cartilage surfaces, decreased and abnormal chondrocyte numbers, and missing menisci in the joint space, as well as fibrous formation on the cartilage surface, worsening fibrosis, and vacuolization being observed. Although meniscus wear was still present due to ligament rupture in the stem cell treatment group, the damage was much less compared with the injury group at the same time point, and the cartilage was neatly arranged, with increased chondrocytes and no severe fibrosis observed (Figure 2F).

Tumor necrosis factor-alpha (TNF-α) and interleukin-6 (IL-6) are inflammatory cytokines closely related to the pathogenesis of osteoarthritis, in which TNF-α can stimulate synovial cells to produce PGE2 and enhance osteochondral destruction while also activating the NF-κB pathway to stimulate various cytokine productions. Moreover, TNF-α and IL-6 jointly participate in exacerbating inflammatory injury and pain transmission in the KOA pathological process. To detect the effects of stem cells on the treatment process of mouse osteoarthritis at different time points, qPCR analysis of the gene expression in mouse osteochondral tissue was performed. Inflammatory factor expression significantly increased in the surgical group compared with the SHAM group. Two weeks post-treatment, no significant difference in TNF-α mRNA expression was noted between the DMM + ADSCs and DMM groups (*p* > 0.05), but IL-6 expression showed a notable difference (*p* < 0.001). By the fourth week post stem cell treatment, the TNF-α and IL-6 expression levels were close to the normal SHAM group levels, significantly differing from those in the DMM group (*p* < 0.05). After treatment cessation, a significant rebound was observed in the 6th week for the TNF-α and IL-6 expression levels in the DMM + ADSCs group, with no significant difference from the DMM group, although they were still lower (*p* > 0.05) (Figure 3B). Simultaneously, the IL-6 content in mouse blood was detected using an ELISA kit, noting a significant increase post-modeling yet revealing a significant decrease in the IL-6 content in the DMM + ADSCs group compared with the DMM group in the 4th week (*p* < 0.01), with no notable difference between the treatment and injury groups at other time points.

An irreversible conversion causing OA occurs when cartilage’s metabolic balance is disrupted, which is closely related to increased activity of extracellular matrix-degrading enzymes matrix metalloproteinase-13 (MMP-13) and a disintegrin and metalloproteinase with thrombospondin 5 (ADAMTS-5). These enzymes can mediate COL-II and Aggrecan degradation, leading to cartilage structural and functional loss. RT-qPCR was used to detect the expression levels of genes such as MMP-13, ADAMTS-5, COL-II, and Aggrecan in animal cartilage tissue, indicating protein synthesis alterations and cellular function in the cartilage injury process. Post modeling, the ADAMTS-5 and MMP-13 expression levels rose. During ADSC treatment, a significant difference in ADAMTS-5 expression was noted between the DMM + ADSCs and DMM groups by the 6th week (*p* < 0.05), and while not statistically different in the 2nd and 4th weeks, a declining expression trend was observable. MMP-13 expression similarly showed significant decreases in the 2nd and 4th weeks (*p* < 0.01). Compared with the SHAM group, there was a significant reduction in the mRNA expression of cartilage formation markers COL-II and Aggrecan throughout the treatment in the modeling group (*p* < 0.05). However, except for the 6th week not showing a significant increase in Aggrecan expression (*p* > 0.05), the COL-II and Aggrecan mRNA expression levels in the DMM + ADSCs group were significantly higher at other nodes compared with the DMM group (*p* < 0.01). Aggrecan expression showed a continuously increasing trend with treatment progression (Figure 3C).

### 3.3. Clinical Manifestations During Dog Modeling and ADSC Treatment

Figure 4 shows the changes during the KOA modeling process in dogs. We used CrCLt surgery for modeling KOA in dogs (Figure 4A). Basic examinations such as heart rate, body temperature, and blood pressure measurement during modeling showed no abnormalities. However, the maximum flexion and extension angle range of the dogs’ knee joints was slightly reduced. As the modeling time increased, the dogs exhibited abnormal movement posture and decreased exercise endurance. Between weeks 4 and 8 of modeling, the dogs gradually showed signs such as alternating leg lameness, excitement, and licking of the hind limbs. By the 8th week of modeling, the lameness score reached grade 2, indicating mild osteoarthritis symptoms. X-ray detection showed a smooth joint surface with no defects before modeling, with normal synovial tissues and a clear infrapatellar fat pad structure. There was no forward sliding of the tibia relative to the femur. After 8 weeks of modeling, no significant defects or osteophyte formations were observed in the dogs’ knee joints, but the tibias moved forward relative to the femurs, indicating CrCL injury and a loss of joint stability. The joint space in the medial region of the dogs’ hind limbs slightly narrowed, and the joint surfaces became rough, while increased density was observed at the joint capsule, suggesting soft tissue swelling. The combined clinical symptoms and imaging findings verified the successful modeling of the KOA model (Figure 4C).

During stem cell treatment, examinations of the heart rate, blood pressure, and temperature of the dogs were within the normal range. Although mild pain manifestations, such as licking of the hind limbs, could still be observed during movement, the joint swelling in the treatment group alleviated after 4 weeks. Measurements of the maximum flexion and extension angles of the treatment and control groups showed better joint extensibility in the treatment group. In terms of the lameness score, the control group dogs showed a progression from intermittent weight-bearing lameness to continuous weight-bearing lameness by the 2nd week, and this persisted until the 8th week of treatment. The treatment group dogs showed intermittent weight-bearing lameness at 4 weeks and continuous weight-bearing lameness at 8 weeks. Throughout the experiment, no dogs exhibited continuous non-weight-bearing lameness. Regarding the activity range, both groups showed reduced exercise endurance, with the treatment group exhibiting mild OA symptoms and the control group exhibiting mild-to-moderate OA symptoms. Pre-sampling X-rays showed no significant osteophyte formation on the joint surfaces for either the control or ADSC treatment groups, and the results were not substantially different from those at 8 weeks of modeling. A lateral view revealed a reduction in the infrapatellar fat pads in both the ADSC treatment group and the control group, indicating wear in the joint (Figure 4D). Dissection showed cartilage wear on the joint surfaces in both the ADSC treatment group and the PBS control group, with the treatment group exhibiting a smaller area of cartilage wear compared with the control group (Figure 4B). The clinical manifestations, imaging, and macroscopic anatomy indicated that stem cell injection positively impacted slowing down OA damage progression.

### 3.4. Comparison with ADSC Treatment of Canine Osteoarthritis

In HE staining, the cartilage surface of the PBS control group appeared incomplete, with a disordered cell arrangement and significant inflammatory infiltration, whereas the ADSC treatment group exhibited less inflammatory cell infiltration compared with the control group (Figure 4E). Toluidine blue staining showed an incomplete cartilage surface in the control group, with lighter staining of the cartilage matrix, blurred boundaries, and disordered cartilage cell arrangement, indicating cartilage degeneration and disordered arrangement of collagen fibers. In the ADSC treatment group, the cartilage surface had mild wear, lighter staining, and relatively irregular cartilage cell arrangement with a stratified cartilage matrix, indicating partial cartilage degeneration (Figure 4F). Safranin O-fast green staining showed that normal cartilage surfaces should be smooth and flat, with neatly arranged cartilage cells, red-stained cartilage layers, four clearly distinguishable structural layers, a clear subchondral bone boundary, and an intact tide mark. Both the control group and the treatment group showed significant surface wear of the cartilage layer, reduced red-stained cartilage layer areas to the point of no coloring, and disordered arrangement of cells in the subchondral bone. Compared with the non-treatment group, the ADSC treatment group exhibited less cartilage wear, better surface integrity, more orderly arrangement of subchondral bone cells, larger red-stained cartilage layer areas, and less severe damage (Figure 4G).

This study detected the expression of inflammatory factor TNF-α and IL-6-related genes in animal cartilage tissues through RT-qPCR. The results showed that 8 weeks post-treatment, IL-6 mRNA expression in the ADSCs group was significantly reduced compared with the control group (*p* < 0.001), and although there was no statistical difference in TNF-α mRNA expression between the two groups, a downward trend was observed in the treatment group (Figure 5A). Compared with the control group, the ADSC treatment group showed significantly lower mRNA expression of ADAMTS-5 and MMP-13 (*p* < 0.05) (Figure 5B) and significantly higher expression of COL-Ⅱ and AGG-2 in cartilage tissue (*p* < 0.001) (Figure 5C).

### 3.5. In Vitro Culture, Proliferation, and Migration Ability Detection of OA Canine Chondrocytes

After isolation, the canine chondrocytes were cultured to the third passage, appearing to be polygon-, triangle-, or irregular spindle-shaped under the microscope, with large, round nuclei centrally located in the cell body, evenly distributed cytoplasm, and dense growth patterns resembling “paving stones” (Figure 5D). After toluidine blue staining, the cultured nuclei appeared dark blue with distinct nucleoli, and the cytoplasm was stained a lighter blue. Based on morphological observation, positive col II immunofluorescence, and positive toluidine blue staining, the results indicated that the extracted cells were articular chondrocytes (Figure 5E,F). Cartilage tissues with canine knee osteoarthritis were extracted from both the ADSC treatment and control groups, and a CCK-8 assay was used to verify whether there were differences in the proliferation ability of chondrocytes from different sources. The results, as shown below (Figure 5G), indicate that the proliferation rate within the detection time for the treatment group was significantly higher than that of the control group (*p* < 0.0001), suggesting that cartilage cells showed better proliferation performance after ADSC injection. Cell migration is a crucial physiological process in living organisms involving organ development, wound healing, and inflammatory responses, among other physiological processes. The cell scratch test can simulate the cell migration process in vivo, studying the mechanisms and regulatory factors of cell migration by observing migration in the scratch area. This experiment compared the migration abilities of chondrocytes under in vitro culture conditions between the KOA damage group and the treatment group, evaluating the effect of ADSC transplantation in enhancing chondrocyte activity and promoting wound healing. At 12 h, 24 h, 48 h, and 72 h after in vitro culture of the P1 generation chondrocytes, the migration rate of chondrocytes in the treatment group was significantly higher than that of the control group (*p* < 0.05) (Figure 5H).

## 4. Discussion

In this study, we evaluated the efficacy of multiple injections of canine ADSCs in counteracting inflammation and joint cartilage damage in a canine OA model. Although ADSCs have been shown to be effective in restoring knee joint function in OA animal models [35], there are no studies indicating changes in inflammatory levels and cartilage repair during multiple injections. Considering animal ethics, we simulated inflammation and tissue changes during treatment using a mouse OA model and verified the feasibility of repeated administration in a canine OA model. In the mouse OA model, cartilage damage was improved, but the anti-inflammatory effect was time-limited. Meanwhile, in the canine OA model, there was a significant therapeutic effect on OA symptoms compared with the control group.

To rule out the influence of cell quality issues, we identified the cell viability, phenotype, and differentiation potential, conforming to international standards. Additionally, the dogs did not show local joint discomfort or pain after each injection. In Cabon’s study [30], nearly half of the joints showed mild-to-moderate discomfort and pain 24 h after the first injection. However, Harman’s study [26] did not have such short-term adverse effects (AEs). The painful joints were mainly in the hips, while our treatment site was the knees. These short-term AEs may be unrelated to the injection procedure and site but related to the number of injections and cell products. Our allogeneic treatment avoided issues like the long expansion time of autologous MSC therapy, secondary injury from tissue collection, and limited quantity [36]. Differences between MSCs may lead to varied therapeutic effects, while allogeneic MSC therapy can harvest MSCs from healthy donors and select the best batch.

The pathogenesis of OA is usually related to changes induced by aging and abnormal mechanical stress in the orthopedic chondrocytic microenvironment [37]. OA is the most common cause of chronic pain in dogs, with an estimated 20% of dogs showing clinical symptoms [22]. Chondrocytes maintain cartilage homeostasis by synthesizing the extracellular matrix (ECM), thus preserving the structural and functional integrity of cartilage. However, in response to aging and abnormal mechanical stress stimuli, chondrocytes lose the ability to maintain cartilage integrity and survival. Not only do stem cells possess differentiation and self-renewal characteristics, directly promoting cartilage repair, but the paracrine system is also an important mechanism for MSC-promoted tissue regeneration [38]. Through multiple injections of ADSCs in OA mouse models, histological studies revealed that ADSC treatment could prevent, stop, or even reverse knee cartilage wear. After treatment stopped at 4 weeks, the cartilage damage rate was still slowed compared with the control group. Similar results were confirmed in the canine OA model, showcasing the role of ADSC treatment in cartilage repair and good performance at the motor level. In vitro experiments also observed enhanced proliferation and migration capabilities of canine chondrocytes, indicating the sustainability of MSC therapy for tissue repair. Restrained by MSC therapy’s intra-articular survival time and the need for long-term OA treatment with continuous administration, studies have attempted to package MSCs with immunoisolation membranes and tissue-engineered products [39,40], combine MSCs with anti-rejection drugs [41], or directly modify MSCs to reduce their immunogenicity [42] and slow down MSC degradation for therapeutic effects. This aligns with our intention to ensure the long-term efficacy of MSC therapy through repeated ADSC injections. In current OA treatment, repeated injections of MSCs show better clinical performance than a single-dose regimen, and there is still no determined ideal source, dose, formulation, or specific set of characteristics in the application of MSC therapy for OA treatment [43].

Inflammation also plays a crucial role in the pathogenesis of OA [44]. Inflammatory factors TNF-α and IL-6 are key mediators of cartilage destruction in OA [45], with studies showing elevated expression levels of TNF-α and IL-6 in the synovial fluid and cartilage tissue of OA patients [46]. IL-6 can upregulate chondrocytic matrix catabolic enzymes, including ADAMTS-5 and MMP-13 [47]. MMP-13 is responsible for the degradation of AGGRECAN and COLLAGEN-II in articular cartilage [48]. Typically, MMP-13 activity is significantly increased in OA animal models [49], and the ADAMTS protein family is also associated with OA cartilage degradation [45]. These detrimental matrix changes further reduce the mechanical integrity and lubrication of cartilage, subsequently accelerating cartilage wear and destruction [50]. The anti-inflammatory properties of stem cells may help reduce OA-related inflammatory processes [51,52] ADSC treatment significantly reduced the elevated expression levels of inflammatory factors TNF-α and IL-6 in mice and weakened the downregulation of AGGRECAN and COLLAGEN-II in cartilage tissue, in addition to the upregulation of ADAMTS-5 and MMP-13. However, the impact on inflammatory factors noticeably weakened after stopping injections at 4 weeks.

In the canine OA model, multiple injections significantly reduced the expression level of IL-6, and although TNF-α showed no statistical difference, it also showed a downward trend. MSCs exhibit anti-inflammatory properties in response to tissue damage or pro-inflammatory states, leading to broad inhibitory effects on the maturation of dendritic cells, macrophages, natural killer (NK) cells, and cytotoxic T lymphocytes [53]. In studies on MSC therapy in mouse OA models, Lchiseki [54] observed that intra-articular cell injection resulted in decreased expression of TNF-α in joint cartilage, as well as reduced levels of COLLAGEN-II and AGGRECAN. This was also confirmed by Saulnier [34] in rabbit OA models. We inferred that injecting ADSCs can effectively promote cartilage repair and extracellular matrix synthesis in OA mice, thus having a therapeutic effect on osteoarthritis. However, the impact on inflammatory factors is time-sensitive, and long-term multiple injection therapy may be necessary to maintain anti-inflammatory actions. Some studies have indicated the potential for combination therapy with MSCs and non-steroidal anti-inflammatory drugs (NSAIDs) [40], which aligns with our objective of achieving an anti-inflammatory effect through multiple injections. The authors of [40] explored the relationship between MSC therapy and AB (Radix Achyranthis Bidentatae) and demonstrated AB’s potential in stimulating mesenchymal stem cell repair for knee osteoarthritis treatment. This indicates that the immune status of the microenvironment is also crucial since MSC differentiation is inhibited by inflammation [55]. Our experiments provide a theoretical basis for maximizing the efficiency of MSC therapy.

OA is a disease intricately linked with the aging process, and its treatment may require long-term sustained administration to maintain a delicate balance between anabolic and catabolic processes in cartilage. With increasing age, the number and severity of comorbidities in OA patients may also increase, possibly adding to treatment complexity [56]. This study still has some limitations, as it did not compare the treatment effects of single and multiple injections in canine OA models. The results are based on an allogeneic treatment model and require further investigation, but they provide new directions for deeper development of MSC therapy in the future. In the quest for MSC therapy for human osteoarthritis, intra-articular injections of MSC have been shown to produce positive results in treatment of the knee in OA phase I and II clinical trials, leading to pain reduction and improved joint and cartilage repair [57]. This approach has gained more potential at the translational level, but more studies with sufficient safety and efficacy are being tested or developed at the clinical level to benefit the majority of OA patients. Our exploration of canine osteoarthritis treatment will provide some data to inform the development and clinical trials of MSC therapies for human osteoarthritis. To fully realize the potential of MSC therapy in the future, maximizing patient benefits and minimizing patient risks, comprehensive adjustments in terms of the number of treatments, immunogenicity, survival rate, potency, disease-specific mechanisms of action, and combination with other drugs are needed to achieve the best therapeutic effects.

## 5. Conclusions

This study demonstrated that multiple intra-articular injections of canine ADSCs could be effective in treating OA symptoms. It demonstrated the modulation of inflammation and matrix renewal-related gene expression in chondrocytes under MSC therapy, suggesting the timeliness of ADSCs’ impact on inflammation and providing the possibility for the combined use of MSC therapy with NSAIDs and other anti-inflammatory drugs.

## Figures and Tables

**Figure 1 cells-14-00323-f001:**
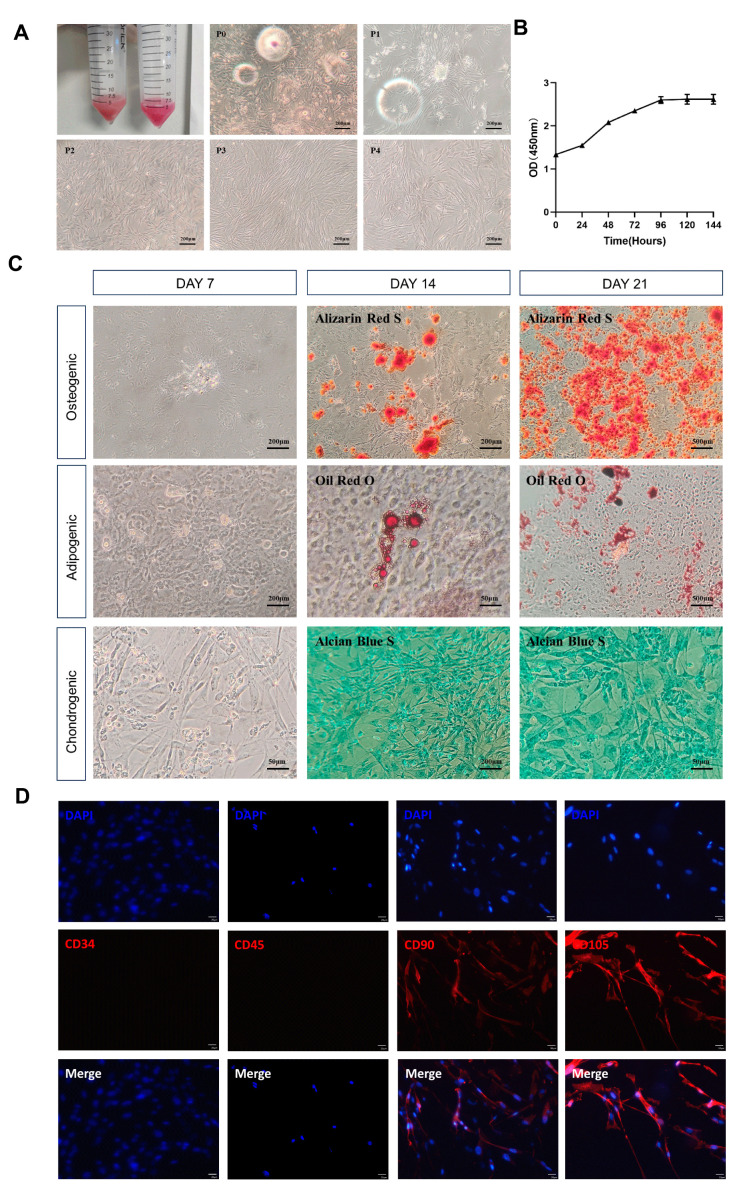
The isolation, culture, and identification of canine ADSCs. (**A**) The morphologies of different passages (P0–P4) of canine ADSCs. (**B**) The cell growth curve of canine ADSCs, (**C**) Analysis of the differentiation capacity of canine ADSCs through lipogenic, osteogenic, and chondrogenic differentiation. (**D**) The immunofluorescence identification of canine ADSC surface markers: CD34, CD45, CD90, and CD105.

**Figure 2 cells-14-00323-f002:**
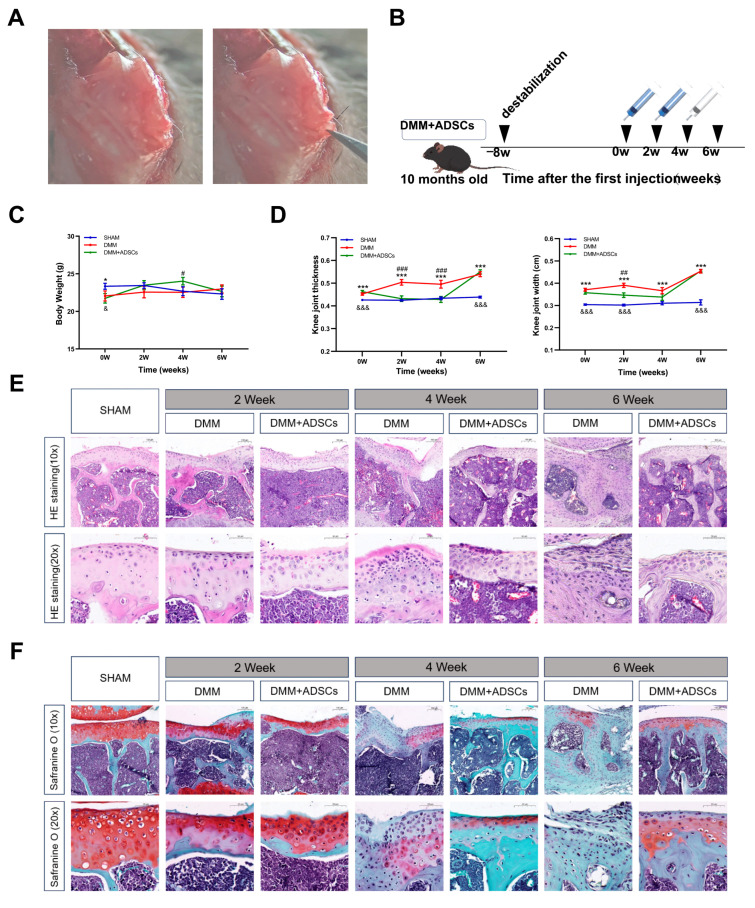
Intra-articular injection of ADSCs to repair damaged cartilage in DMM mice. (**A**) The modeling process of the DMM mice model, with the meniscus indicated by arrows. (**B**) Schematic diagram of injection treatment time schedule. (**C**) Changes in the body weight of mice during the course of the treatment period. (**D**) Changes in thickness and width of the knee joints during the course of treatment of the mice. (**E**) HE staining of articular cartilage. (**F**) Tomato red-O-solid green staining of articular cartilage. * meanings SHAM compared DMM, & meaning SHAM compared DMM + ADSCs, # meanings SHAM compared DMM. * ^# &^
*p* ≤ 0.05. ^##^. *p* ≤ 0.01. *** ^### &&&^
*p* ≤ 0.001.

**Figure 3 cells-14-00323-f003:**
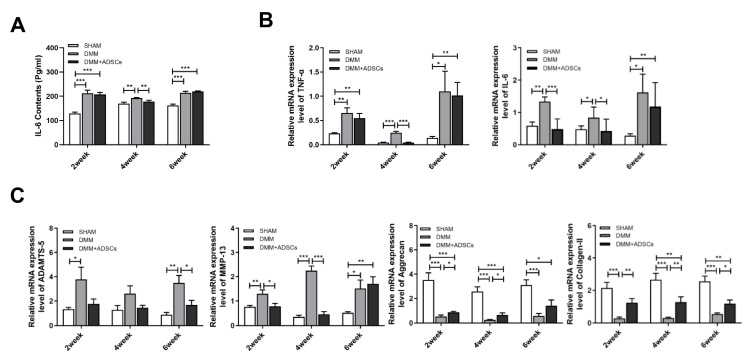
Analysis of inflammation and cartilage repair-related indexes during treatment in DMM mice. (**A**) Serum IL-6 levels during treatment. (**B**) Real-time PCR analysis of the expression of TNF-α and IL-6 in cartilage samples during treatment. (**C**) Real-time PCR analysis of cartilage samples for chondrogenesis-related genes ADAMTS-5 and MMP-13 during treatment, as well as Aggrecan and Collagen-II mRNA expression. * *p* ≤ 0.05. ** *p* ≤ 0.01. *** *p* ≤ 0.001.

**Figure 4 cells-14-00323-f004:**
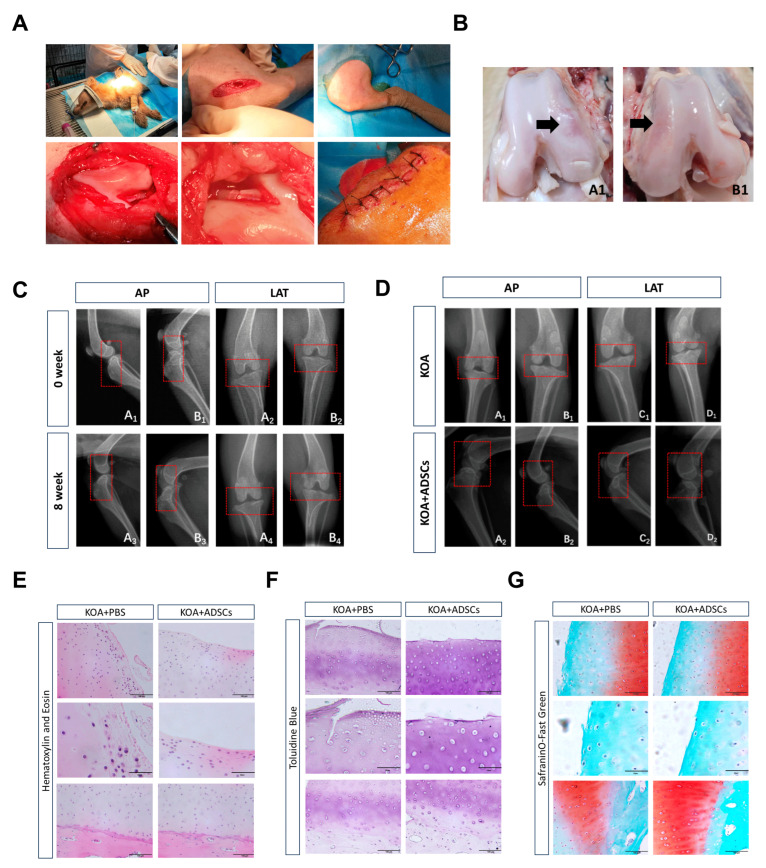
Intra-articular injection of ADSCs to repair damaged cartilage in KOA canines. (**A**) Modeling process of KOA canine model. (**B**) Gross observation of articular cartilage, where A1 is the ADSC treatment group, B1 is the control group, and the abrasion point is indicated by an arrow. (**C**) X-ray examination of the affected limb before and after modeling of the canines. (**D**) X-ray examination of the affected limb before and after treatment of the canines. Joints are highlighted in the red box (**E**) HE articular cartilage staining. (**F**) Toluidine blue articular cartilage staining. (**G**) Tomatoxylin-O-fixed green articular cartilage staining.

**Figure 5 cells-14-00323-f005:**
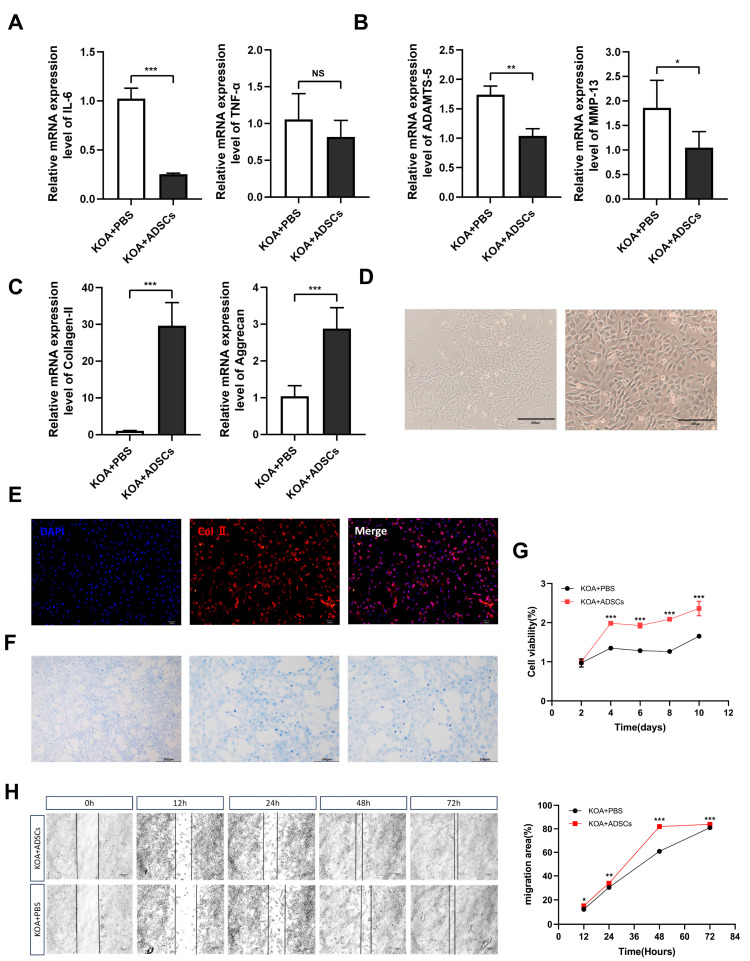
Changes in inflammation and cartilage repair-related indexes during treatment in KOA dogs. (**A**) Real-time PCR analysis of mRNA expression of inflammatory factors IL-1β and IL-6 in cartilage samples during treatment. (**B**) Real-time PCR analysis of mRNA expression of chondrogenesis-related genes ADAMTS-5 and MMP-13 in cartilage samples during treatment. (**C**) Real-time PCR analysis of mRNA expression of chondrogenesis-related genes Collagen-II and Aggrecan in cartilage samples during treatment. (**D**) Chondrocyte morphology. (**E**) Immunofluorescence identification of canine chondrocytes’ Col-II. (**F**) Chondrocyte toluidine blue staining. (**G**) CCK-8 detection of value-added canine chondrocyte cells. (**H**) Scratch assay for detection of canine chondrocyte cell migration ability. NS meaning no significance.* *p* ≤ 0.05. ** *p* ≤ 0.01. *** *p* ≤ 0.001.

## Data Availability

The original contributions presented in this study are included in the article. Further inquiries can be directed to the corresponding author(s).

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
