# Peer review of "Multiple Intra-Articular Injections of Adipose-Derived Mesenchymal Stem Cells for Canine Osteoarthritis Treatment"

_cells, 2025, doi:10.3390/cells14050323_

Round 1
Reviewer 1 Report
Comments and Suggestions for Authors
General characteristics and evaluation of the reviewed article:
The article addresses the important topic of osteoarthrosis (OA) therapy in dogs using multiple injections of adipose-derived mesenchymal stem cells (ADSCs). The studies presented include a mouse and a dog model, providing convincing evidence for the efficacy of ADSCs in OA therapy. However, the article contains some methodological and factual shortcomings that may affect its perception and scientific value.
The article is well written and based on sound research methods. However, several aspects need to be improved to increase clarity, quality of interpretation of results and potential application in clinical practice. I provide detailed comments below.
Minor comments:
Please provide a more comprehensive discussion of osteoarthritis in the first paragraph. This will create a stronger introduction to the topic and emphasize the importance of the issue. The incidence of osteoarthritis is affected by various factors such as occupation, sports participation, musculoskeletal injuries, obesity, and gender. Information about these factors, along with relevant literature, should be included in the first paragraph of the introduction. I recommend adding the following references to this section:
https://doi.org/10.3390/healthcare12161648
DOI: 10.1056/NEJMcp1903768
The article does not compare the efficacy of multiple injections with a single dose, making it impossible to assess whether repeated administrations are actually more effective. The authors should add a control group in which ADSCs are administered only once, and compare the results with the multiple injection group.
The study ends relatively soon after the last injection, and OA is a chronic disease. It is not known how long the therapeutic effect lasts. The follow-up period should be extended to at least 6-12 months to assess the durability of the treatment effect. Please elaborate more on the limitations and include my recommendations in your plan for future work.
The article is mainly based on regenerative and anti-inflammatory effects, but does not analyze the mechanism of cell action within the joint. It would be worth expanding the study to assess the presence and survival of ADSCs in the joint (e.g., by labeling the cells) and to analyze the secretion of cytokines and growth factors.
The article lacks precise information on the number of ADSC cells administered and the volume of injection. It is necessary to clarify the number of cells per kg of body weight and the volume of each injection, which will allow better replication of the results.
Whether repeated injections of ADSCs can induce an immune response in dogs has not been analyzed. It would be worthwhile to add studies assessing the immune response, such as serum levels of pro-inflammatory cytokines (TNF-α, IL-1β).
The authors do not refer to clinical trials of ADSCs in OA in humans, which could lend credence to the translational nature of the results. It would be worthwhile to add a comparison to the clinical trials and discuss how these results may affect future treatment of OA in humans.
I congratulate the authors of an interesting study, I believe that after the appropriate amendments are made, the work will be further processed and published. I wish you further success!
Reviewer 2 Report
Comments and Suggestions for Authors
General comments
This is a well-written and well-reasoned manuscript presenting the results of a well-designed and reliably performed experiment. My comments are therefore editorial in nature and I am happy to recommend this manuscript for publication after minor revision.
Specific comments
L 10-L 11 Replace the abbreviation expansion to the first reference in the text.
L 13 Expand all abbreviations in the first place where are refered in the text.
L 14-L19 Consider replacing this part by more specific data of M&M and results from this study. Avoid generalization and speculation.
L 25-26 Consider replacing „in canine and human populations” with „in human and animals populations, including canines”
L 32 Remove additional dot.
L 48-49 Avoid repetitions (see L 36-36).
L 91-96 As this experiment was conduct on the single joint (knee joint), consider adding the joint name to the aim of the study.
L 129 Was only one mouse used in the experiment?
L 146 Expand all abbreviations, also KOA, in the first place where are refered in the text.
L 160 Details of the X-ray system used to take the images. Who graded the X-rays and how did they grade the severity of OA?
L 166 Specify exactly how long after the first surgical intervention in the knee joint the tissue was collected. How many dogs were used in the experiment? Specify dogs’ breed, age, sex, and knee joint healthy status before surgical intervention. Was it the same period of time for each dog? I did not find any information about euthanasia of dogs before collecting the material.
L 199 Whether and how the data distribution was examined before choosing a nonparametric test.
L 195 Expand all abbreviations, also TNF, in the first place where are refered in the text.
L 323 Expand all abbreviations, also MMP and ADAMTS. See comment above.
L 357 In Figure 4, the radiographs are too small to be diagnostic. Because I appreciate the layout of the entire figure, consider adding each of the subfigures (B, C, D, E, F, and G) as a separate subfigure in the supplementary materials so that interested readers can take a closer look at the lesion/signs you are presenting.
L 463 and L 466 I did not find in the M&M section how joint discomfort and pain were assessed. I am also interested in how you differentiated lameness, pain and discomfort resulting from damage to the cranial cruciate ligament from lameness, pain and discomfort associated with intra-articular injection.
L 475 chronic orthopedic pain, as it is not general true
L 527 Expend AN and AB
L 527 Expend NSAID
L 546 – L 547 I can not agree with these two sentences. As no clinical sympthoms of OA vere evaluated and graded; and no comparison with single injection was clearly presented.
Reviewer 3 Report
Comments and Suggestions for Authors
Thank you for the interesting study on multiple application of ad-MSC in dogs.
L1 You assessed the applicability both in mice and dogs. The title only refers to dogs. Please explain and adapt.
L20 ADMSC could be effective in treating OA symptoms
L32 The references 6,7 and 8 are not correctly used as they are concerning exosomes.
Many of your references seem to have shifted in the text and are not used consecutively from 1. to 52. Please check and correct accordingly
L35 Ref 9 us concerning adipocytes. Ref 10 is related to synovium Ref 11 is missing as is Ref 20. Ref 12 is mice not dental pulp. Ref 14 is related to placenta-MSC. Same problem for 21, 26, 27 and 30. Please recheck the entire text.
L51 Ref 11 wrong position
L69 Ref 22 + 23 wrong position
L133 I presume between patellar ligament and medial collateral ligament
L135 I presume of the medial meniscus. Please correct. This description is not in line with L246 where you transect the medial meniscus and medial collateral. Please correct
L139 suturing the joint capsule.....
L139 after instrumentaion of the mice
L142 instrumented knee joint
L146 KOA I presume knee OA.
L161 before instrumentation of the dogs
L165 Collection of all dog and mice joints. What was collected and how exactly processed?
L172 Scratch test was performed on chondrcytes harvested from the joints. Please clarify
L 242 - 250 This section seems to belong to M & M and not results. Please correct
L386 3.3. should be 3.4.
L470 Ref 54 is not existing
L527 AN[50] ????? Please correct
L546 could be effective
